# OpenReview forum: "Failures Pave the Way: Enhancing Large Language Models through Tuning-free Rule Accumulation"
_EMNLP/2023/Conference — EMNLP 2023 Main_

### Official Review · Reviewer_Ucpj · 2023-07-24

**Typos Grammar Style And Presentation Improvements:** L210, "with the processes (b) and (h)…
**Soundness:** 4

**Excitement:**

4: Strong: This paper deepens the understanding of some phenomenon or lowers the barriers to an existing research direction.

**Paper Topic And Main Contributions:**

This paper proposes a tuning-free rule accumulation framework to prevent frozen LLMs repeating similar mistakes. Specifically, the framework guides LLMs to generate rules for subsequent deployment when the model encounters mistakes. Strategies for LLMs to autonomously manage and maintain the rule collection is also designed to improve the efficiency of the rule collection. Experiments show that the framework significantly enhances performance on several scenarios.

**Questions For The Authors:**

How does TRAN complement CoT? Based on my understanding, TRAN + zero-shot CoT is concat rules before "Let's think step-by-step", and TRAN + few-shot CoT is rules + ICL examples with CoT

**Reasons To Accept:**

1. This paper proposes a tuning-free rule accumulation framework to prevent frozen LLMs repeating similar mistakes. The rule collection construction and maintenance process proposed in this paper has been carefully designed. The proposed method can effectively avoid frozen LLMs repeating similar mistakes in real-world scenarios.
2. The experiments and analysis in this paper are in-depth and comprehensive. The method is evaluated on both online and conventional train-test setting. Cross-domain and counterfactural scenarios are also considered.
3. This paper is well-written and easy to understand.

**Reasons To Reject:**

I don't find any reasons to reject this paper. If I had to point a limitation, it would be that the framework is only evaluated on multi-choice QA and text classification tasks. It is possible that this framework could also demonstrate potential effectiveness in other tasks, such as decision-making tasks like ALFWorld.

**Reproducibility:**

4: Could mostly reproduce the results, but there may be some variation because of sample variance or minor variations in their interpretation of the protocol or method.

**Reviewer Confidence:**

4: Quite sure. I tried to check the important points carefully. It's unlikely, though conceivable, that I missed something that should affect my ratings.

---

> ### Author Rebuttal · Authors · 2023-08-29
>
> Thank you for the constructive suggestions and comments.
>
> ### Q1: How does TRAN complement CoT? Based on my understanding, TRAN + zero-shot CoT is concat rules before "Let's think step-by-step", and TRAN + few-shot CoT is rules + ICL examples with CoT
>
> Yes, in the few-shot setting, the template is ***"Rules + ICL examples"***. On the other hand, in the zero-shot setting, we concat the rules before the question prompt instead of before 'Let's think step-by-step'. The template of TRAN + zero-shot CoT is ***"Rules + Question + Answer + 'Let's think step-by-step'"***. We acknowledge the need to provide clarity, and we will include examples of the prompt design in the next version of our paper.

---

### Official Review · Reviewer_ziX2 · 2023-08-06

**Soundness:** 4

**Excitement:**

4: Strong: This paper deepens the understanding of some phenomenon or lowers the barriers to an existing research direction.

**Paper Topic And Main Contributions:**

This paper presents a novel Tuning-free Rule Accumulation (TRAN) framework, which harnesses LLMs to summarize rules from the mistakes they makes, and then uses these rules to guide future predictions. The method involves prompting a LLM to generate rules from single/multiple incorrect cases, validating and deduplicating the generated rules, and subsequently leveraging these rules for making predictions. The experiments demonstrate the strong performance of the TRAN framework on multi-choice question answering and text classification tasks.


**Questions For The Authors:**

1. Is the proposed method fully automatic? How generalizable would it be on other NLP tasks? Would involving some human intervention to guide or validate the rules be helpful?
2. Can you discuss possible directions to further advance this research?

**Reasons To Accept:**

1. This paper demonstrates that a LLM can successfully abstract rules from its previous mistakes and leverage these rules to avoid making similar mistakes. This is a novel idea for guiding LLMs to correct their predictions.
2. The method allows for self-adaptation of black-box LLMs in an online learning fashion. It can iteratively accumulate rules based on observed mistakes in the streaming data, thus constantly improving the model.
3. The rule maintenance method further guarantees the collection of rules are accurate, effective, and do not grow excessively.


**Reasons To Reject:**

1. The experimental setup lacks clarity. In Table 1, is it performing online rule accumulation on the test set? Additionally, how does the ordering of the examples affect the performance? Moreover, in Tables 1&2, it is unclear how the CoT and few-shot baselines are implemented. It is unclear what is the differences between the experiments in Table 5 and Table 1. Does it mean Table 5 uses the training set and Table 1 does not? If so, why the results using the training set are lower than those without? Overall, the lack of clarity in the experimental setup might affect my judgment of the soundness of the results.
2. While it is intriguing that the model can summarize rules by itself, it would be beneficial to see more examples and analysis of how frequently the model generates reasonable rules and whether there are any incorrect rules that could potentially impact the model negatively.
3. SALAM is an important baseline compared in the experiments. It is not clearly described what is this baseline and how the proposed method differs from it.


**Reproducibility:**

4: Could mostly reproduce the results, but there may be some variation because of sample variance or minor variations in their interpretation of the protocol or method.

**Reviewer Confidence:**

4: Quite sure. I tried to check the important points carefully. It's unlikely, though conceivable, that I missed something that should affect my ratings.

---

> ### Author Rebuttal · Authors · 2023-08-29
>
> Thank you for the valuable comments and suggestions. We have carefully revised our paper.
>
> ## Q1: The experimental setup lacks clarity. In Table 1, is it performing online rule accumulation on the test set? Additionally, how does the ordering of the examples affect the performance? Moreover, in Tables 1&2, it is unclear how the CoT and few-shot baselines are implemented. It is unclear what is the differences between the experiments in Table 5 and Table 1. Does it mean Table 5 uses the training set and Table 1 does not? If so, why the results using the training set are lower than those without? Overall, the lack of clarity in the experimental setup might affect my judgment of the soundness of the results.
>
> ### Q1.1: In Table 1, is it performing online rule accumulation on the test set?
> Yes, online rule accumulation is performed on the test set.
>
> ### Q1.2: How does the ordering of the examples affect the performance?
>
> We shuffle the data by three different seeds and report the results on three datasets. As shown below, our method consistently demonstrates competent performance across three orders, in comparison to the default sequencing. Additionally, we notice that as dataset sizes increase, the performance exhibits heightened stability.  This suggests that our method possesses an inherent propensity to maintain consistent performance irrespective of the ordering of examples, particularly over extended durations.
>
> | dataset | size | seed = 0 | seed = 1 | seed = 2 | average | default (Table 1) |
> | :-- | --: | :---: | :---: | :---: | :---: | :---: |
> | **BBQ Religion** | 680 | 90.74\%  | 89.12\% | 89.26\% | **89.71\%** | **89.71\%** |
> | **BBQ Disability** | 1,156 | 86.77\% | 89.71\% | 90.31\% | **88.93\%** | 88.15\% |
> | **BBQ Nationality** | 1,880 | 94.52\% | 94.41\% | 94.41\% | 94.45\% | **94.73\%** |
>
> ### Q1.3: How are CoT and few-shot baselines implemented?
>
> We implement few-shot baselines in the template of ***"Task Description + ICL Examples + Answer: "***. We have provided examples of few-shot prompt designs in Table 8-11, which can be found in the Appendix. In the case of the zero-shot CoT baseline, we adhere to the template ***"Task Description + Question + Answer: Let's think step by step."*** as outlined by Kojima et al. (2022)[1]. Furthermore, for the Auto-CoT baseline, we implement the template proposed by Zhang et al. (2022)[2], presenting demonstrations prior to the input question. We will incorporate concrete examples of the CoT baselines in the next version of our paper.
>
> ### Q1.4: What are the differences between the experiments in Table 5 & Table 1? Does it mean Table 5 uses the training set and Table 1 does not?
>
> Yes, the experiments in Table 5 utilize the training set, whereas those in Table 1 do not.
>
> As mentioned in Section 4, we adopt the train-test split setting by Wang and Li (2023)[3] for better comparison, which entails selecting 250 samples randomly and dividing the dataset using a 0.8/0.2 split ratio. Notably, the rule collection remains frozen during testing.
>
> In contrast, the experiments in Table 1 are conducted on the entire dataset, which is substantially larger than the 250 samples mentioned earlier, as depicted in Table 7. By progressively accumulating rules, our TRAN method effectively leverages prior knowledge to enhance subsequent generations. Consequently, over extended periods, it's plausible that TRAN could demonstrate improved performance.
>
> ## Q2: While it is intriguing that the model can summarize rules by itself, it would be beneficial to see more examples and analysis of how frequently the model generates reasonable rules and whether there are any incorrect rules that could potentially impact the model negatively.
>
> As mentioned in Section 2.3, we only retain rules that can rectify the current mistake, ensuring the majority of rules are reasonable and effective. We have further reinforced this assertion by showcasing the most and second most frequently employed rules for each dataset in Table 19.
>
> However, it is essential to acknowledge that during our experiments, we encountered instances where certain rules were incorrect and could potentially mislead the model. The most prevalent instance of this occurrence was observed in cases of over-correction. For example, in the TweetEval Irony dataset, we notice that the rule `If the review does not contain the hashtag #NOT or any other indicators of denial commonly used on social media to emphasize non-irony, then the review is not ironic` was utilized more than 50 times. While this rule ostensibly captures a form of irony involving sarcasm through opposites, it is evidently an incorrect rule.
>
> In essence, our method entails the model's involvement in both rule generation and evaluation, inevitably leading to the existence of incorrect rules. Nevertheless, based on our current experiments and observations, the prevalence and utilization of such incorrect rules appear to be marginal. And we intend to enrich our work with additional comparative examples and in-depth analyses in the future work.
>
> ## Q3: SALAM is an important baseline compared in the experiments. It is not clearly described what is this baseline and how the proposed method differs from it.
>
> Generally speaking, our approach aims at enhancing the self-alignment capabilities of LLMs in a streaming setting. In contrast, SALAM is designed to assist LLMs to reflect and refine the initial answer with the mistakes in the training phase.
>
> To elaborate, SALAM adopts an auxiliary assistant model to provide feedback while we utilize the main model across all stages of the process. Moreover, for each input question, we directly utilize the accumulated rules to guide the generation. On the other hand, SALAM retrieves similar mistakes to construct specific guidelines to refine the current answer.
>
> ## Q4: Is the proposed method fully automatic? How generalizable would it be on other NLP tasks? Would involving some human intervention to guide or validate the rules be helpful?
>
> Yes, our method is fully automatic.
>
> As also mentioned by Reviewer 17gG, we further conduct experiments on the **Dyck Language** task from **Big-Bench Hard** (Suzgun et al. 2022)[4], which is predicting sequences of closing parentheses of Dyck-4 words without last closing parentheses. One example input is `Complete the rest of the sequence, making sure that the parentheses are closed properly. Input: \[ \{ \[` and the oracle answer is `\] \} \]`.
>
> The comparative results presented below underscore the substantial advancements achieved by our approach compared to the zero-shot baseline. Notably, utilizing zero-shot CoT leads to performance degradation, in line with findings in [4]. In essence, our approach demonstrates promising potential for QA tasks. However, we recognize a pivotal distinction in the Dyck Languages task: its inherent presence of concrete laws, which distinguishes it from conventional long-form QA tasks. The refinement of rule structures and construction, to enable the extension of our framework to universal tasks, is left as an important future work.
>
> Furthermore, as mentioned in Q2, our method inherently encounters the challenge of incorrect rules. Thus, it is definitely helpful to involve human intervention to write and validate rules. According to the following results, the inclusion of a manually crafted rule (`+ Human`) yields performance enhancements surpassing both the zero-shot and our approaches. We intend to explore better interaction with humans in the future.
>
> | Dataset | Zero-Shot | Zero-Shot CoT| Zero-Shot + Human | Ours | Ours + Human |
> | :-------- | :--------: | :--------: | :--------: | :--------: | :--------: |
> | BBH Dyck Languages | 39.6\% | 36.4\% | 40.8\% | **44.4\%** | **45.2\%** |
>
> ## Q5: Can you discuss possible directions to further advance this research?
>
> Here are several possible directions to advance this research.
>
> - First, as mentioned in your insightful comments, our current approach is fully automatic, which faces the challenges of uncontrollable rules. To enhance real-world applicability, it is essential to explore approaches that effectively interact with humans.
> - Moreover, in this work, we guide LLMs to generate rules in an intuitive approach. One possible direction lies in incorporating other well-designed reasoning methods. By incorporating these methods, we can potentially generate rules that are more reasonable, versatile, and adaptable.
> - Furthermore, we intend to evaluate and advance our approach in environments where the preference is dynamic, which reflects complex real-world scenarios. This is a crucial step toward real-world deployment.
>
> In summary, this paper is a preliminary exploration into harnessing LLMs for micro alignment without the need for extensive fine-tuning. It remains largely unexplored how to effectively adapt LLMs to dynamic environments better.
>
> ### References
>
> [1] Takeshi Kojima, Shixiang Shane Gu, Machel Reid, Yutaka Matsuo, and Yusuke Iwasawa. 2022. Large language models are zero-shot reasoners. In ICML 2022 Workshop on Knowledge Retrieval and Language Models.
> [2] Zhuosheng Zhang, Aston Zhang, Mu Li, and Alex Smola. 2022. Automatic chain of thought prompting in large language models.
> [3] Danqing Wang and Lei Li. 2023. Learn from mistakes through cooperative interaction with study assistant.
> [4] Mirac Suzgun, Nathan Scales, Nathanael Schärli, Sebastian Gehrmann,
> Yi Tay, Hyung Won Chung, Aakanksha Chowdhery, Quoc V. Le, Ed H. Chi, Denny Zhou, and Jason Wei. 2022. Challenging BIG-Bench tasks and whether chain-of-thought can solve them.

---

### Official Review · Reviewer_17gG · 2023-08-06

**Soundness:** 4

**Excitement:**

4: Strong: This paper deepens the understanding of some phenomenon or lowers the barriers to an existing research direction.

**Paper Topic And Main Contributions:**

This paper introduces an approach to dynamically accumulating a set of LLM-generated "rules", which are custom "if/then" instructions that can be retrieved and used to improve LLM performance at some task. The system is fed examples (e.g. a multiple-choice question) one at a time. When it makes a mistake, it first retrieves similar mistakes it has made, and then an LLM is prompted to generate a natural-language "rule" in "if/then" format that should correct this type of mistake.

If the rule is effective (the LLM doesn't make the mistake when provided with the rule), it is added to the rule collection. Then, when similar examples are seen in the future, the rule is retrieved from the collection and added to the prompt. If it makes a mistake again, the process is repeated. When new rules are added to the rule collection, the system retrieves similar rules and prompts an LLM to check for any that are redundant or contradictory.

In this way, the system continually creates and refines a set of rules that improve the performance on some task without any parameter tuning. The authors compare to a variety of zero-shot and few-shot baselines, showing very strong metric gains in the former and mild gains in the latter.

**Questions For The Authors:**

**Q1**: What considerations would be necessary in order to expand to a task where mistakes cannot be programmatically determined? For example, in long-form QA? For sake of example, let's assume there is a human annotator who would flag a mistaken response, but no other information. Would you expect this framework to work well on such a task out of the box, or what else would you need to look out for?

**Q2**: Did the system seem to generally succeed in retrieving relevant rules from the collection? I'm curious whether your BM25 retrieval system did the job. I could see it struggling to retrieve a rule like "If a judgment is based on a stereotype, then it is not true" from an example like "Who probably has an okay memory, my daughter or my grandpa?"

**Reasons To Accept:**

- The method integrates nicely into other prompt engineering ideas like CoT prompting etc.
- Benchmarks against the similar SALAM method
- Shows dramatic performance against against previous methods in the zero-shot setting
- Very forward-thinking approach in the way that it uses LLMs in multiple different ways to create an effective system
- Their analysis on domain generalization is quite insightful for understanding the merit and possible failure modes of the method

**Reasons To Reject:**

Tasks evaluated are limited to classification and multiple choice QA, presumably so that they it can be programmatically determined whether a model has made a mistake without human input. It would be more compelling if it included a task or two with longer form LLM responses such as long-form QA or dialogue. Such a setting could make it much harder for the model to automatically pinpoint the cause of a mistake and generate an effective rule.

It would have also been nice to see some numbers on the hit rate of your rule retrieval system.

**Reproducibility:**

3: Could reproduce the results with some difficulty. The settings of parameters are underspecified or subjectively determined; the training/evaluation data are not widely available.

**Reviewer Confidence:**

3: Pretty sure, but there's a chance I missed something. Although I have a good feel for this area in general, I did not carefully check the paper's details, e.g., the math, experimental design, or novelty.

---

> ### Author Rebuttal · Authors · 2023-08-29
>
> Thank you for the helpful comments and suggestions.
>
> ### Q1: What considerations would be necessary in order to expand to a task where mistakes cannot be programmatically determined? For example, in long-form QA? For the sake of example, let's assume there is a human annotator who would flag a mistaken response, but no other information. Would you expect this framework to work well on such a task out of the box, or what else would you need to look out for?
>
> To further evaluate our method, we conduct experiments on the **Dyck Language** task from **Big-Bench Hard** (Suzgun et al. 2022)[1], where the model is required to complete the sequences of the closing parentheses of a Dyck-4 word without its last few closing parentheses. One example question is `Complete the rest of the sequence, making sure that the parentheses are closed properly. Input: \[ \{ \[` and the corresponding target answer is `\] \} \]`.
>
> According to the comparative results shown below, our approach gains substantial improvement over the zero-shot baseline. Also, we notice that using zero-shot CoT degrades the performance, in line with the results reported in [1]. In summary, our approach exhibits potential for QA tasks. Nevertheless, we acknowledge a distinct characteristic of the Dyck Languages task is the presence of concrete laws that distinguish it from conventional long-form QA tasks. We leave advancing the rule structures and construction to extend our framework to universal tasks in future work.
>
> | Dataset | Zero-Shot | Zero-Shot CoT| Ours |
> | :-------- | :--------: | :--------: | :--------: |
> | BBH Dyck Languages | 39.6\% | 36.4\% | **44.4\%** |
>
> ### Q2: Did the system seem to generally succeed in retrieving relevant rules from the collection? I'm curious whether your BM25 retrieval system did the job. I could see it struggling to retrieve a rule like "If a judgment is based on a stereotype, then it is not true" from an example like "Who probably has an okay memory, my daughter or my grandpa?"
>
> To evaluate the retrieval performance, we analyze the results on three datasets: BBQ Religion, Disability, and Nationality, all evaluated under the train-test setting. In each dataset, we identified approximately 80\%, 100\%, and, 90\% of mistakes that exhibit the potential for correction through effective rules. Within these subsets of errors, our retrieval system achieved success in rectifying around 75\%, 66.7\%, and, 66.7\% of the identified mistakes. For the instances where our system's retrieval did not yield successful results, we found that approximately 16.3\%, 4.7\%, and, 10.5\% of the effective rules from the complete **rule collection** were applicable. Notably, for each input, multiple effective rules are often applicable, and we select the top three most relevant rules. Overall, our BM25 retrieval system demonstrates a satisfactory success rate.
>
> We recognize the need for a comprehensive analysis of our retrieval system and the potential for incorporating more refined retrieval strategies. These aspects are envisioned as part of our future work.
>
> ### References
>
> [1] Mirac Suzgun, Nathan Scales, Nathanael Schärli, Sebastian Gehrmann,
> Yi Tay, Hyung Won Chung, Aakanksha Chowdhery, Quoc V. Le, Ed H. Chi, Denny Zhou, and Jason Wei. 2022. Challenging BIG-Bench tasks and whether chain-of-thought can solve them.

---

### Meta-Review · Area_Chair_LzXX · 2023-09-18

**Recommendation:** 3

**Metareview:**

Proposes Tuning-free Rule Accumulation (TRAN), a prompting framework for an LLM (gpt-3.5-turbo) to generate a self-managed set of rules that are used in future prompts to prevent similar mistakes. Relevant rules are retrieved via BM25.

It appears to improve performance over baselines on Q/A and text classification tasks. Some experiments are done in an 'online learning' fashion on the test set which is reflective of production use cases, making it hard to assess the lift over baselines in Table 1. It's recommended to clarify this. In Table 5, a fairer comparison is made with SALAM, however, comparing with other baselines would be more informative, especially given Table 1 suggests SALAM is not universally better than others and it is a new, less common baseline.

## Pros
- Mostly reproducible -- prompt templates provided
- Good results on some QA and classification tasks (see caveat below)

## Cons
- Online rule accumulation is performed on the test set so it's not quite fair to other baselines that do not see the test set in Table 1 experiments.
- Tested on limited types of tasks, unclear how to generalize to other generative tasks.

---

### Decision · Program_Chairs · 2023-10-07

**Decision:**

Accept-Main

**Comment:**

Proposes Tuning-free Rule Accumulation (TRAN), a prompting framework for an LLM (gpt-3.5-turbo) to generate a self-managed set of rules that are used in future prompts to prevent similar mistakes. Relevant rules are retrieved via BM25.

It appears to improve performance over baselines on Q/A and text classification tasks. Some experiments are done in an 'online learning' fashion on the test set which is reflective of production use cases, making it hard to assess the lift over baselines in Table 1. It's recommended to clarify this. In Table 5, a fairer comparison is made with SALAM, however, comparing with other baselines would be more informative, especially given Table 1 suggests SALAM is not universally better than others and it is a new, less common baseline.

## Pros
- Mostly reproducible -- prompt templates provided
- Good results on some QA and classification tasks (see caveat below)

## Cons
- Online rule accumulation is performed on the test set so it's not quite fair to other baselines that do not see the test set in Table 1 experiments.
- Tested on limited types of tasks, unclear how to generalize to other generative tasks.